# Response of oat morphologies, root exudates, and rhizosphere fungal communities to amendments in a saline-alkaline environment

**Peina Lu[1,2], Tony Yang[2], Lijun Li[1], Baoping Zhao[1], Jinghui Liu[1]***

**1** College of Agronomy, Inner Mongolia Agricultural University, Hohhot, Inner Mongolia, P.R. China, **2** Swift Current Research and Development Centre, Agriculture and Agri-Food Canada, Swift Current, Canada

\* cauljh@aliyun.com

**Data Availability Statement:** All relevant data are within the manuscript and its Supporting Information files. All raw sequencing data used in this article were uploaded to NCBI Sequence Read

## Abstract

The application of organic amendments to saline-alkaline soil has been recommended as an agricultural strategy to improve crop productivity and soil health. However, there has been limited research on how organic soil amendment strategies affect the health of oats and their associated rhizosphere fungal communities in saline-alkaline conditions. Thus, the objectives of this study were to understand the effects of oat cultivars with contrasting saline-alkaline tolerances and different amendments on plant morphologies, root exudates (soluble sugars and organic acids), and rhizosphere fungal communities in a saline-alkaline environment. Experiments were conducted on a saline-alkaline tolerant cultivar, Baiyan2, and a saline-alkaline sensitive cultivar, Caoyou1, under four different organic amendment strategies: 1. control (no amendment application), 2. bio-fertilizer application, 3. rotten straw application, and 4. a co-application of bio-fertilizer and rotten straw. Results showed that plant morphological characters of Baiyan2 were better than Caoyou1, and that soluble sugar and organic acid levels in the rhizosphere of Baiyan2 were significantly lower than Caoyou1. Compared to the control, oat root and plant development was significantly improved by the combined bio-fertilizer and rotten straw amendment. Bio-fertilizer application promoted malic and citric acid levels, contributing to a higher total organic acid level, and significantly increased the abundance of *Rhizopus arrhizus* and decreased the abundance of the fungal pathogens *Alternaria*, *Cladosporium*, *Sarocladium* and *Heydenia* of *Ascomycota* in both oat cultivars. All amendment treatments containing rotten straw, except the combined amendment in Baiyan2, significantly increased the relative abundance of *Ascomycota* (specifically *Gibberella*, *Talaromyces*, *Fusarium*, and *Bipolaris*) and decreased the relative abundance of *R. arrhizus* by reducing soluble sugar and organic acid levels. For the combined amendment in Baiyan2, there were no significant changes in *Gibberella* and *Rhizopus* between the control and amendment treatment. Our results suggest that co-application of bio-fertilizer and rotten straw, combined with a tolerant oat cultivar, is an effective method to increase crop productivity and enhance soil health in a saline-alkaline environment.

Archive (SRA) under bioproject accession number PRJNA670290.

**Funding:** This work was funded by Scientific and Technological Innovation Team of Inner Mongolia; Multi-grain Engineering and Technology Center of Inner Mongolia; Talent Introduction Project of Inner Mongolia Agricultural University (NDYB2018-29) and MOE-AAFC PhD Research program (201808150135).

**Competing interests:** The authors have declared that no competing interests exist.

## Introduction

Due to increased demands for high quality food products, sustainable agroecosystems, and usable farmland worldwide, the possibility of using saline-alkaline land for agriculture is of interest to researchers [1–4]. Organic soil amendment strategies, such as such as bio-fertilizer and decomposed straw application, have been shown to improve the quality of saline-alkaline soils [5–8]. These amendments function by increasing available nutrients, modifying soil structure, and promoting plant growth [5–8]. Bio-fertilizers are substances containing living microorganisms [9] that can colonize crops and promote growth in several ways [10]. Rotten straw application can enhance root growth by increasing soil nutrients and optimizing soil compaction [11, 12]. Several studies have reported that the co-application of bio-fertilizers and organic manuers like rotten straw can efficiently modify soils, facilitate development of crop roots, and enhance plant growth in agricultural ecosystems [6, 13, 14].

Oat (*Avena nuda* L.) has been cultivated worldwide [15] and is used as a phytoremediation crop to ameliorate saline-alkaline soil, especially in arid and semi-arid areas [16]. Oat cultivars with contrasting saline-alkaline tolerances [17, 18] have differing productivities in the same saline-alkaline ecosystems [16], making tolerant cultivar selection a valuable strategy for increasing productivity. Different cultivars may also influence the surrounding soil microorganisms, as crop species and genotype have been shown to shape the soil microbial community [19]. This is significant because the soil microbial community can influence plant growth [19]. One mechanism by which plants alter microbial communities is through root exudations. Root exudates (e.g. sugars and organic acids) from plants exposed to various biotic and abiotic factors (i.e. salt, pH) [20–22] and are closely related to plant survival under poor nutrient conditions [23–25]. Specifically, many studies have shown that the richness of rhizosphere microbial communities can be enhanced by increased root exudations, such as: organic acids and carbohydrate [23, 26–28]. Fungal communities in particular are often affected by certain crop species [29] and selectively regulated by certain root exudates [30].

Multiple interactions occur between organic soil amendments, root exudates, and the microbial community. Some researchers have reported that bio-fertilizer significantly increases soil organic acid contents [10, 31, 32], because the bio-remediation contains many beneficial microbes, such as *Rhizopus arrhizus*, known to produce malic, citric, fumaric acid in saline-alkaline soils [32–34]. Other research has suggested that organic management practices alter the soil microbial community by altering soil nutrient conditions [5]. Bio-fertilizers may act directly on soil nutrients, promoting plant growth, which allows soil microorganisms to receive more carbohydrates and organic acids from the host plant [35]. Rotten straw application may also affect the microbial community. Maarastawi et al. [36] observed that the assimilation of root exudates by rhizosphere organisms is reduced in the presence of straw because microorganisms preferentially use straw as a carbon source over root exudates.

There has been limited research on how organic soil amendment strategies affect the health of oats and their associated rhizosphere fungal communities in saline-alkaline conditions. Thus, the objectives of this study were 1) to examine the effects of organic soil amendments on plant morphologies, root exudates, and associated rhizosphere fungal communities in two oat cultivars, and 2) to define the relationships between these variables in a saline-alkaline environment. Our results provide evidence that plant morphology and root exudate contents are negatively correlated and vary between crop cultivars under different amendments, introducing a possible technique for evaluating the relationships between root exudates and rhizosphere fungal communities in the improvement process of saline-alkaline soils.

## Materials and methods

### Study site and soil amendment compositions

This two-year field experiment was conducted at the Tumote Zuoqi Hailiu village (111°22′30″ E, 40°41′30″ N) of Hohhot, Inner Mongolia, China from 2016 to 2017. The study site has a mean annual temperature of 13.2°C and precipitation of 410 mm. Local soils are classified as saline-alkaline, with a pH of 9.14, EC of 1.55 mS cm$^{-1}$, 63.11 mg kg$^{-1}$ of available N, 15.71 mg kg$^{-1}$ of available P, 171.33 mg kg$^{-1}$ of available K, 0.53 g kg$^{-1}$ total N, 1.66 g kg$^{-1}$ total P, 0.53 g kg$^{-1}$ total K, and 13.32 g kg$^{-1}$ of organic matter, as measured at the time of plot establishment in May 2016.

The bio-fertilizer used in this study had a pH of 6.85, EC of 125.93 μS cm$^{-1}$, 532.42 mg kg$^{-1}$ of available N, 166.52 mg kg$^{-1}$ of available P, 222.79 mg kg$^{-1}$ of available K, 84.67 g kg$^{-1}$ total N, 0.16 g kg$^{-1}$ total P, 9.33 g kg$^{-1}$ total K, and 372.60 g kg$^{-1}$ of organic matter and the bio-fertilizer with mixture of bacteria (62.77% *Proteobacteria*, 15.76% *Firmicutes*, 18.36% *Bacteroidetes* and so on). The bio-fertilizer treatment involved application at 150 kg ha$^{-1}$ yr$^{-1}$ during oat seeding in both 2016 and 2017.

Rotten straw was generated from 5 cm long sections of corn straw that were fermented without oxygen for 8 months (from September to April) directly preceding each experiment. Rotten straw had an average pH of 6.58, EC of 2.95 μS cm$^{-1}$, 362.06 mg kg$^{-1}$ of available N, 169.70 mg kg$^{-1}$ available of P, 345.39 mg kg$^{-1}$ of available K, 14.06 g kg$^{-1}$ total N, 0.18 g kg$^{-1}$ total P, 7.99 g kg$^{-1}$ total K and 552.48 g kg$^{-1}$ of organic matter, rotten straw main including the bacteria (78.61% *Proteobacteria*, 20.95% *Firmicutes* and so on) and fungi (98.19% *Ascomycota*). The rotten straw treatment consisted of rotten straw applied evenly on the soil surface at 12 000 kg ha$^{-1}$ yr$^{-1}$ and mixed with top 15 cm of soil using a rotary tiller prior to seeding in both 2016 and 2017.

Diammonium phosphate (DAP: 18-46-0) was applied at a rate of 150 kg ha$^{-1}$ yr$^{-1}$ in all plots during oat seeding, which main consider that the lower soil nutrients in this field.

### Study experimental design

This study consisted of 8 treatments (2 oat cultivars and 4 amendment treatments) arranged in a randomized complete block design, each with three replications. The oat cultivars selected were Caoyou1, a saline-alkaline sensitive cultivar, and Baiyan2, a saline-alkaline tolerant cultivar [16]. Oats were seeded at a rate of 150 kg ha$^{-1}$ yr$^{-1}$ with 25 cm row spacing in the springs of both 2016 and 2017. In the results, A and B represent the Caoyou1 and Baiyan2 oat cultivars, respectively. CK (A1 and B1) is the negative control; F (A2 and B2) is the bio-fertilizer treatment; R (A3 and B3) is the rotten straw treatment; and RF (A4 and B4) is the bio-fertilizer plus rotten straw treatment.

### Sampling and measurements

**Plant sampling and measurements.** Oat plants were removed from three randomly selected 0.25 m×0.25 m subplots of each plot in September 2017 after 60 days of soil amendments. We harvested the aboveground plants, then dug up the crop roots and shook off loose bulk soil. Any adhered soil was washed off under running tap water in lab. After cleaning, the morphological structure of each root sample was assessed using a root scanner (Microtek ScanMaker i800 Plus, Zhejiang Top Instrument Co., Ltd., China) and analysis software (LA-S Analysis, Hangzhou Wanshen Detection Technology Co., Ltd., China). Fresh biomass of each shoot and root sample was measured. Samples were then dried at 70°C until constant weight

and dry biomass was recorded (48 h) with an analytical balance (JY10002, Shanghai Hengping Instrument and Meter Factory, China).

**Rhizosphere soil sampling and measurements.** Rhizosphere soil samples were collected at the same time and from the same 0.25 m×0.25 m subplots as the plant samples. Loose soil on the oat roots from each subplot was shaken off by hand in the field and rhizosphere soil within 2 mm of each sample was gently collected in a Ziploc© bag. Each sample was sieved through a 2 mm mesh screen and stored at -80°C for root exudate collection and DNA extraction.

**Root exudate collection.** Root exudates were collected from 10 g subsamples of each rhizosphere soil sample. These subsamples were dissolved in 100 mL of deionized water in a 250 mL Erlenmeyer flask, and shook on an orbital shaker (HY-4 speed multi-purpose shaker, Jiangsu Dongpeng Instrument Manufacturing Co., Ltd., China) for one hour at 200 rpm. The resulting solutions were centrifuged (L530R Low Speed Refrigerated Tied Centrifuge, Hunan Xiangyi Centrifuge Instrument Co., Ltd., China) at 50 xg for 5 minutes to compact the soil. Supernatants of each extraction sample were transferred to a filter apparatus and passed through a 0.45 μm membrane filter. All extraction samples were stored at -20°C for root exudate analysis.

**Root exudate analysis.** The root exudates analyzed were organic acids and soluble sugar. For organic acids, the extraction samples were collected through a membrane filter (water system, pore size 0.3 μm). Organic acid concentration was then determined by high performance liquid chromatography (HPLC) (LC-20AT, Shimadzu, Japan) with the following running conditions: UV monitor (SPD-20A), Atlantis of TMdC18 column (4.6 mm×250.0 mm, 5 μm), 1 mL min$^{-1}$ flow rate, 25°C column temperature, and 20 μL injection volume. Mobile phase A contained 0.01 mol L$^{-1}$ Na$_2$HPO$_4$ (pH = 2.7, 1 mol L$^{-1}$ phosphoric acid regulation), mobile phase B contained acetonitrile, and mobile phase C contained ultra-pure water. UV absorbance was detected at 210 nm. Organic acids from standards and samples dissolved in HPLC-grade methanol were injected into the sample loop and the mean peak areas of individual compounds were taken for quantification. Organic acids present in the sample were identified by comparing their retention time to the organic acid standards of oxalic acid (3.655 min), tartaric acid (4.262 min), formic acid (4.644 min), malic acid (5.477 min), ascorbic acid (6.729 min), acetic acid (7.791 min), citric acid (10.206 min), fumaric acid (11.618 min) and succinic acid (12.552 min; Sigma-Aldrich, St. Louis, MO, USA). Soluble sugar content was determined using the anthrone method [37]. All analysis of organic acids were performed by Inner Mongolia Agriculture, Animal Husbandry, Fishery and Biology Experiment Research Centre (Inner Mongolia, China).

**DNA extraction, PCR amplification, and pyrosequencing.** Total raw DNA was extracted from 0.5 g of fresh soil using the PowerSoil DNA Isolation Kit (Mo Bio Laboratories, Carlsbad, CA, USA). Concentration and purity of the raw DNA samples were tested with a 1% agarose gel (1% AGE, 100 V/40 min), and qualified samples were stored at -80°C for further analysis. Each DNA sample was diluted with nuclease-free water to 1 ng μL$^{-1}$ for PCR amplification. The ITS1 region of the fungal internal transcribed spacer (ITS) was amplified using the specific primers ITS5-1737F (5′–GGAAGTAAAAGTCGTAACAAGG–3′) and ITS2-2043R (5′–GCTGCGTTCTTCATCGATGC–3′), in accordance with Huang et al. [38]. To ensure efficiency and accuracy, all amplification of fungal ITS sequences was performed using the primers described above with a 6-nt barcode, Phusion High-Fidelity PCR Master Mix with GC Buffer (New England Biolabs, UK), and template DNA. Each PCR reaction was performed in a 30 μL-volume system containing 15 μL of Phusion Master Mix (2×), 3 μL of 6 μM forward and reverse primers, 10 μL of template DNA and 2 μL of nuclease-free water. Thermocycler conditions consisted of an initial denaturation at 98°C for 1 min, 35 cycles of 98°C for 10 s, 50°C for 30 s, 72°C for 30 s, and a final extension at 72°C for 5 min. PCR products were purified using

the QIAquick PCR Purification Kit (QIAGEN, Germany) and subjected to quantitation using a Qubit1 2.0 Fluorometer (Invitrogen, USA).

Purified amplicons were pooled in equimolar concentrations and used for library construction with the NEB Next® Ultra™ DNA Library Preparation Kit for Illumina (NEB, USA) following the manufacturer's recommendations. Index codes were added to the constructed libraries. Final DNA quality was assessed with a Qubit 2.0 Fluorometer (Invitrogen, USA) and Agilent Bioanalyses 2100 system. Libraries were sequenced on an Illumina HiSeq2500 platform, with 250 bp paired-end reads generated by Novogene Bioinformatics Technology Co. Ltd, Beijing, China. The raw sequencing data were submitted to NCBI Sequence Read Archive (SRA) under bioproject accession number PRJNA670290.

Paired-end reads were divided into sample libraries according to unique barcodes, shortened by removing the PCR primer sequence and barcodes, and merged using FLASH [39]. Raw tags were filtered to produce high-quality clean tags refer to Qiime [40, 41]. Then chimeric sequences were removed using the UCHIME Algorithm [42] based on the reference Unite Database [43] to produce the final effective tags. Sequences were clustered into OTUs (Operational Taxonomic Units) at 97% identity using UPARSE [44]. The representative sequence for each OTU was classified by Qiime software based on the Unite Database [43].

## Statistical analysis

The effects of amendment treatments on plant morphologies, root exudates, and rhizosphere fungal communities were evaluated using the general linear models (GLM) procedure and significant differences among means were separated using Fisher's least significant difference (F-LSD) at 5% level by SAS 9.0. Qiime software (Version 1.9.1) was used to calculate the diversity indices (observed richness, chao1 index, and Shannon index). Principal component analysis (PCA) [45] and Adonis analysis based on Bray-Curtis dissimilarity matrices were used to test the effects of treatments on microbial community using the "vegan", "pairwise", "ade4" and "ggplot2" packages in R. Redundancy analysis (RDA) [46] was used to evaluate the relationship between plant morphologies, root exudates, and fungal communities using the "rda ()" function in R [47, 48]. In addition, Spearman correlation analysis and heatmaps of correlation were used to analyze the relationships between plant morphologies and root exudates and fungal communities in the different treatments.

## Results

### Oat growth

We observed that oat cultivar selection had a significant effect on all plant morphologies except root dry biomass and root length (Table 1). Specifically, Baiyan2 had better morphological characters than Caoyou1. Compared to the control, organic amendment strategies (treatments F, R, and RF) significantly improved all plant morphological parameters except root dry biomass (Table 1). In particular, shoot and root biomass, root length, root volume, root surface area, and root diameter of both oat cultivars were significantly higher in the RF treatment (Table 1). Root fresh biomass of Caoyou1, and root biomass of Baiyan2, were higher in the bio-fertilizer (F) treatment than the control (Fig 1). Root fresh biomass and root volume of Caoyou1, and root volume of Baiyan2, were significantly higher in both treatments containing rotten straw (R and RF; Fig 1).

### Oat root exudates

This study found strong oat cultivar × amendment interactions for soluble sugar and organic acid levels in the rhizospheres of Caoyou1 and Baiyan2 (Table 2). In general, Caoyou1 had

**Table 1. Effects of soil amendments on plants morphologies of Caoyou1 and Baiyan2 oat cultivars.**

| Treatments | Shoot fresh biomass | Shoot dry biomass | Root fresh biomass | Root dry biomass | Root length | Root volume | Root surface area | Root diameter |
|---|---|---|---|---|---|---|---|---|
| | g plant⁻¹ | | | | cm | cm³ | cm² | cm |
| **Cultivar (C)** | | | | | | | | |
| **Caoyou1** | 6.87b [b] | 1.9b | 1.32b | 0.35a | 444.06a | 43.28b | 287.23b | 2.28b |
| **Baiyan2** | 10.03a | 3.42a | 1.63a | 0.40a | 495.87a | 55.95a | 411.73a | 2.85a |
| **Amendment (M)** [a] | | | | | | | | |
| **CK** | 6.68b | 2.11b | 1.09b | 0.21c | 303.37b | 28.98b | 218.37c | 1.88b |
| **F** | 8.68ab | 3.00a | 1.51a | 0.44b | 507.07a | 44.21b | 337.48bc | 2.26b |
| **R** | 9.01a | 2.57a | 1.58a | 0.37ab | 518.06a | 60.31a | 370.9ab | 2.54b |
| **RF** | 9.44a | 2.96a | 1.72a | 0.49a | 551.35a | 64.96a | 471.17a | 3.57a |
| **ANOVA table (LSD protected, $P \leq 0.05$)** | | | | | | | | |
| **C** | < .0001 | < .0001 | 0.003 | 0.100 | 0.074 | 0.002 | 0.003 | 0.011 |
| **M** | 0.050 | 0.087 | 0.001 | < .0001 | < .0001 | < .0001 | 0.002 | < .0001 |
| **C*M** | 0.229 | 0.615 | 0.006 | 0.910 | 0.921 | 0.006 | 0.187 | 0.129 |

[a] CK is the negative control; F is the bio-fertilizer treatment; R is the rotten straw treatment; and RF is the bio-fertilizer and rotten straw treatment.

[b] Values are represented as means, and the letters in the cultivar (C) and amendment (M) sections represent a significant difference between values at a $p < 0.05$ level.

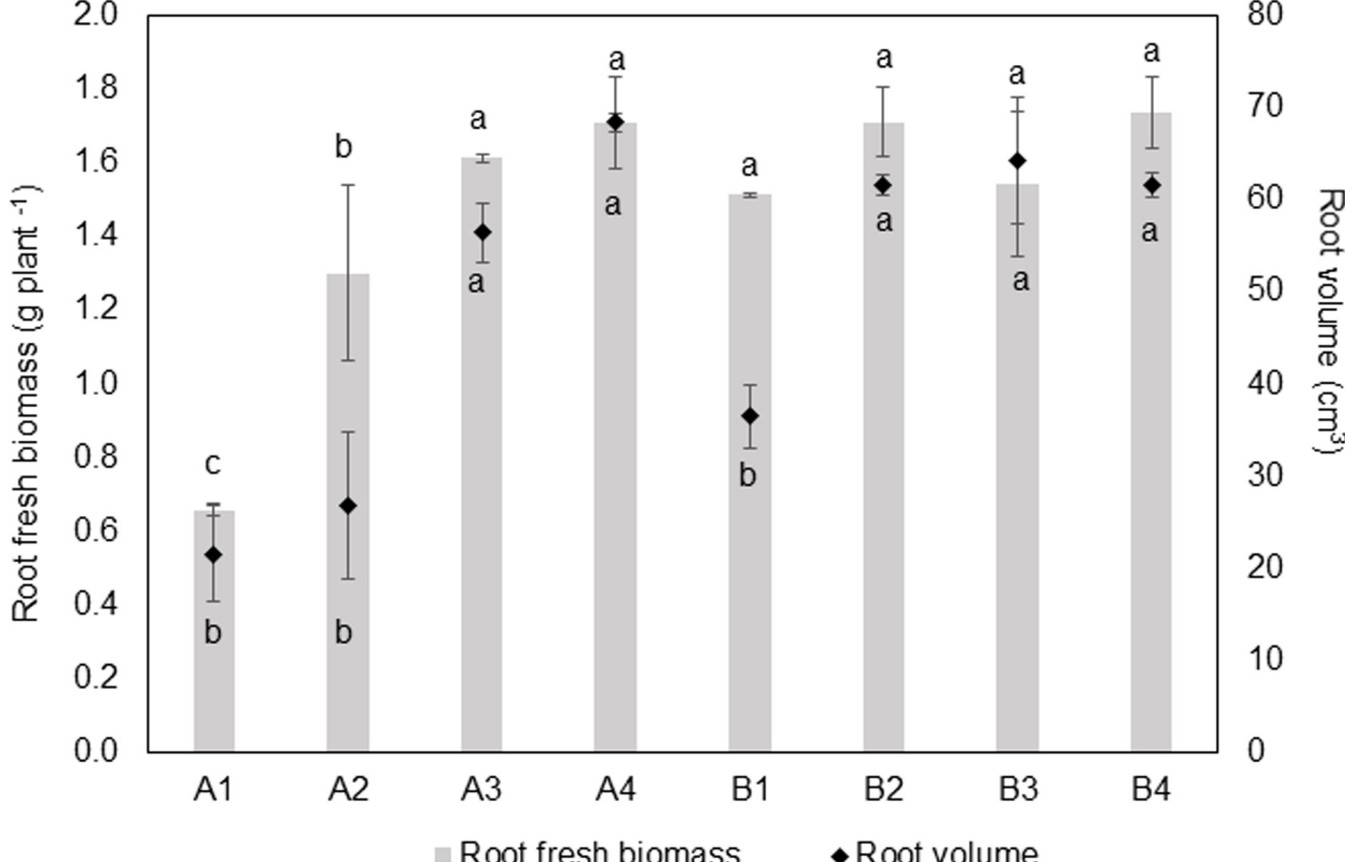

**Fig 1.** Effects of soil amendments on root volume and biomass of Caoyou1 (A) and Baiyan2 (B) oat cultivars. A1 and B1 represent the negative control; A2 and B2 represent the bio-fertilizer treatment; A3 and B3 represent the rotten straw treatment; and A4 and B4 represent the bio-fertilizer and rotten straw treatment.

**Table 2. Effects of soil amendments on soluble sugar and organic acid levels in the rhizosphere of Caoyou1 and Baiyan2 oat cultivars (ug·g dry soil$^{-1}$).**

| Treatment | Soluble sugar | Oxalic | Tartaric | Formic | Malic | Ascorbic | Acetic | Citric | Fumaric | Succinic | Total organic acid |
|---|---|---|---|---|---|---|---|---|---|---|---|
| **Cultivar (C) [a]** | | | | | | | | | | | |
| **Caoyou1(A)** | 272.46a[c] | 35.62a | 97.53a | 94.37a | 318.06a | 9.64a | 64.39a | 58.74a | 5.65b | 38.83a | 722.83a |
| **Baiyan2(B)** | 218.63b | 28.14b | 62.75b | 84.7b | 106.31b | 8.36a | 20.61b | 55.28b | 6.92a | 22.37b | 395.44b |
| **Amendment (M) [b]** | | | | | | | | | | | |
| **CK** | 333.03a | 25.72ab | 128.63a | 157.04a | 239.38a | 16.17a | 77.86a | 46.02b | 5.65a | 44.68a | 741.14a |
| **F** | 211.13ab | 41.68ab | 56.02bc | 52.4b | 396.93a | 5.98b | 45.38a | 165.36a | 7.27a | 35.03a | 806.04a |
| **R** | 123.6b | 12.45b | 91.5b | 74.15b | 122.69b | 6.17b | 5.16b | 9.54c | 6.09a | 5.88b | 333.65b |
| **RF** | 314.42a | 47.66a | 44.4c | 74.53b | 89.72b | 7.7b | 41.61a | 7.14c | 6.13a | 36.82a | 355.7b |
| **C*M [b]** | | | | | | | | | | | |
| **A1** | 465.55b | 38.70b | 150.04a | 176.71a | 337.00b | 21.64a | 147.01a | 83.29c | 4.34d | 52.84b | 1011.56a |
| **A2** | 330.05c | 71.70a | 39.35f | 7.38g | 636.65a | 5.34d | 82.00b | 137.54b | 5.66bcd | 67.44a | 1053.07a |
| **A3** | 149.59e | 14.62c | 120.52b | 87.73d | 179.52c | 5.61d | 4.19e | 7.81d | 7.26ab | 5.94e | 433.21cd |
| **A4** | 144.65e | 17.45c | 80.19d | 105.65c | 119.06f | 5.99d | 24.35d | 6.32d | 5.35cd | 29.12d | 393.48d |
| **B1** | 200.51d | 12.74c | 107.21c | 137.38b | 141.77e | 10.69b | 8.70e | 8.75d | 6.95bc | 36.53cd | 470.72c |
| **B2** | 92.21f | 11.66c | 72.68de | 97.43cd | 157.22d | 6.62cd | 8.75e | 193.18a | 8.88a | 2.61e | 559.02b |
| **B3** | 97.62f | 10.28c | 62.48e | 60.57e | 65.86g | 6.73cd | 6.13e | 11.27d | 4.93d | 5.83e | 234.09f |
| **B4** | 484.18a | 77.87a | 8.61g | 43.41f | 60.37g | 9.40bc | 58.86c | 7.95d | 6.92bc | 44.53cd | 317.93e |
| **ANOVA table (LSD protected, $P \leq 0.05$)** | | | | | | | | | | | |
| **C** | < .0001 | < .0001 | < .0001 | < .0001 | < .0001 | 0.112 | < .0001 | 0.035 | 0.009 | < .0001 | < .0001 |
| **M** | < .0001 | < .0001 | < .0001 | < .0001 | < .0001 | < .0001 | < .0001 | < .0001 | 0.081 | < .0001 | < .0001 |
| **C*M** | < .0001 | < .0001 | < .0001 | < .0001 | < .0001 | < .0001 | < .0001 | < .0001 | 0.002 | < .0001 | < .0001 |

[a] A and B represent the Caoyou1 and Baiyan2 oat cultivars, respectively.

[b] CK (A1 and B1) is the negative control; F (A2 and B2) is the bio-fertilizer treatment; R (A3 and B3) is the rotten straw treatment; and RF (A4 and B4) is the bio-fertilizer and rotten straw treatment.

[c] Values are represented as means, and the letters in the cultivar (C), amendment (M) and C*M sections represent a significant difference between values at a $p<0.05$ level.

higher levels of soluble sugar and all tested organic acids (except fumaric acid) than Baiyan2 (Table 2). Additionally, soluble sugar levels were significantly decreased by most organic amendment treatments (F, R, and RF). However, soluble sugar levels were elevated in the B4 treatment (Table 2). Bio-fertilizer application (F) significantly increased citric acid and total organic acid levels, and significantly decreased tartaric, formic and ascorbic acid levels in both oat cultivars (Table 2). All tested organic acids except fumaric acid were significantly reduced in treatments containing rotten straw (R and RF; Table 2) in Caoyou1. However, some acids, i.e. oxalic, acetic, and succinic acid, were significantly elevated in the B4 treatment (Table 2).

## Oat rhizosphere fungal diversity

PCA results showed that the rhizosphere fungal community was affected by both oat cultivar selection and amendment application (Fig 2). For cultivars, observed richness was higher in Baiyan2 than in Caoyou1 (Table 3). For amendments, the rhizosphere fungal community differed significantly between the bio-fertilizer treatment (F) and the negative control (CK), and between the treatments containing rotten straw (R and RF) and the treatments without rotten straw (CK and F; Fig 2). The R and RF treatments significantly increased the observed richness, chao1 index, and Shannon index of the fungal community, whereas the F treatment significantly decreased the Shannon index (Table 3).

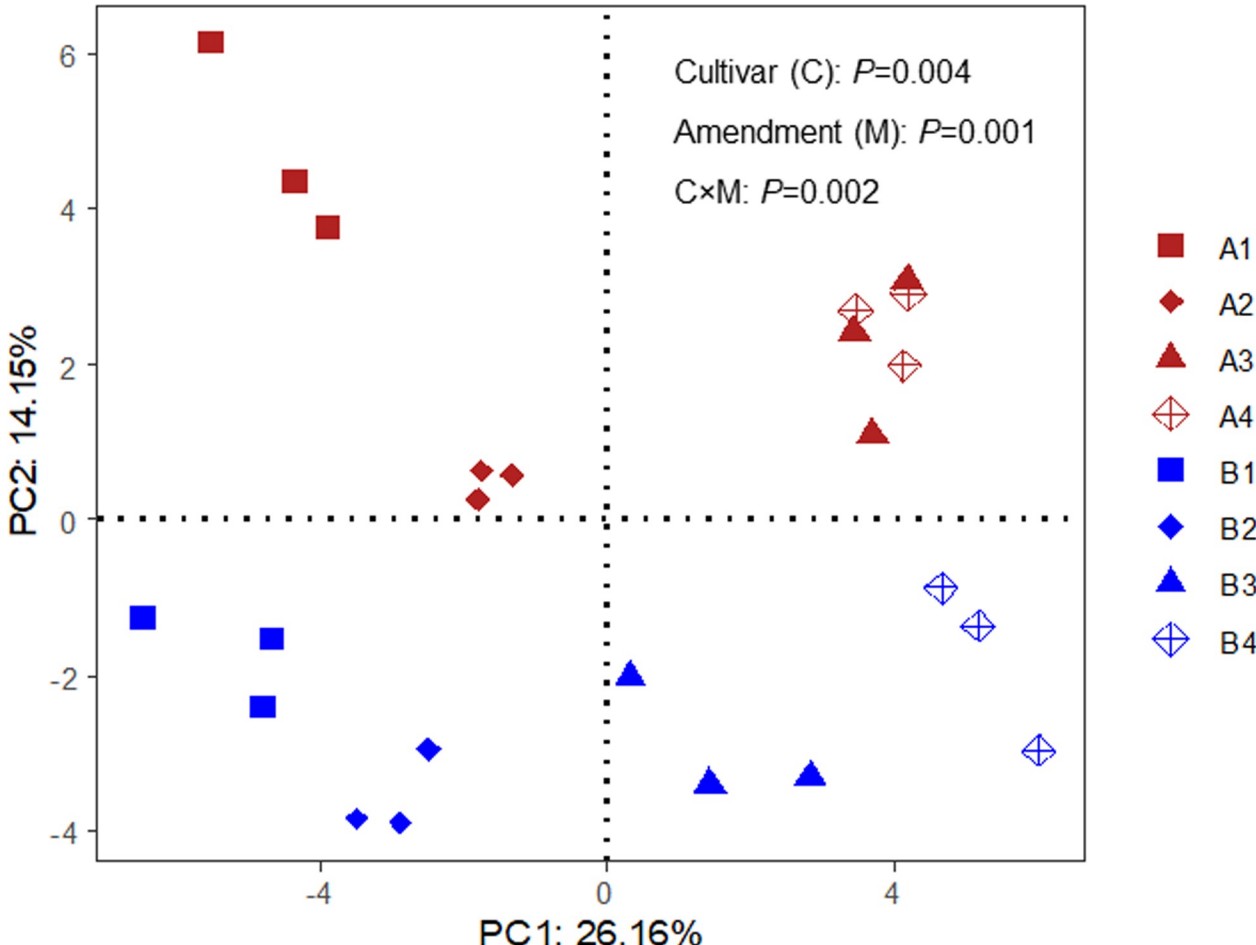

**Fig 2. Principal component analysis (PCA) plot showing the fungal communities detected in the rhizosphere of two oat cultivars under different soil amendment treatments.** A and B represent the Caoyou1 and Baiyan2 oat cultivars, respectively. A1 and B1 represent the negative control; A2 and B2 represent the bio-fertilizer treatment; A3 and B3 represent the rotten straw treatment; and A4 and B4 represent the bio-fertilizer and rotten straw treatment.

## Oat rhizosphere fungal community composition

The top three identified fungal phyla in this study were *Ascomycota* (58.89%), *Zygomycota* (27.72%) and *Basidiomycota* (13.07%), and the top three identified fungal genera were *Rhizopus* (26.42%), *Gibberella* (23.10%), and *Alternaria* (6.85%) (Fig 3A and 3B). Oat cultivar significantly affected the relative abundances of fungal genera (S2 Table). In particular, the relative abundances of *Gibberella*, *Talaromyces*, *Fusarium*, *Heydenia*, and *Bipolaris* were higher in rhizosphere of Baiyan2 than Caoyou1 (Fig 3B and S2 Table).

The effects of amendment and cultivar × amendment interactions on the fungal community were significant (S1–S3 Tables). At the phylum level, the A2, B2, and B3 treatments significantly decreased the proportion of *Ascomycota*, and significantly increased the proportion of *Zygomycota*, as compared to the control (Fig 3A and S1 Table). On the contrary, the A3 and A4 treatments had the inverse effect of these phyla, increasing *Ascomycota* and decreasing *Zygomycota* (Fig 3A and S1 Table). There were no significant differences in fungal phyla between the B1 and B4 treatments (Fig 3A and S1 Table).

At the genus and species level, all organic amendment treatments (F, R, and RF) significantly decreased the proportions of *Alternaria* and *Cladosporium* in both oat cultivars

**Table 3. Effects of soil amendments on rhizosphere fungal α-diversity indexes of Caoyou1 and Baiyan2 oat cultivars.**

| Treatments | Observed richness | Chao1 | Shannon |
|---|---|---|---|
| **Cultivar (C)** | | | |
| **Caoyou1** | 279.17b [b] | 320.21a | 3.5a |
| **Baiyan2** | 300.67a | 343.99a | 3.63a |
| **Amendment (M) [a]** | | | |
| **CK** | 274.5b | 307.9b | 3.65a |
| **F** | 268.5b | 306.42b | 3.12b |
| **R** | 304.33a | 343.23a | 3.7a |
| **RF** | 312.33a | 370.83a | 3.78a |
| **ANOVA table (LSD protected, $P \leq 0.05$)** | | | |
| **C** | 0.010 | 0.064 | 0.104 |
| **M** | 0.001 | 0.005 | < .0001 |
| **C*M** | 0.077 | 0.160 | 0.400 |

[a] CK is the negative control; F is the bio-fertilizer treatment; R is the rotten straw treatment; and RF is the bio-fertilizer and rotten straw treatment.

[b] Values are represented as means, and the letters in the cultivar (C) and amendment (M) sections represent a significant difference between values at a $p<0.05$ level.

compared to control (Fig 3B, S2 and S3 Tables). Bio-fertilizer application (F) significantly increased the proportion of *Rhizopus* (*R. arrhizus*) in both cultivars, whereas rotten straw application (treatments R and RF) significantly decreased the relative abundance of *Rhizopus* (*R. arrhizus*) in Caoyou1. The R and RF treatments also significantly decreased the proportion of *Heydenia* in Baiyan2, and significantly increased the proportion of *Gibberella* in Caoyou1 as well as *Talaromyces*, *Fusarium* and *Bipolaris* in Baiyan2 compared to control (Fig 3B and S2

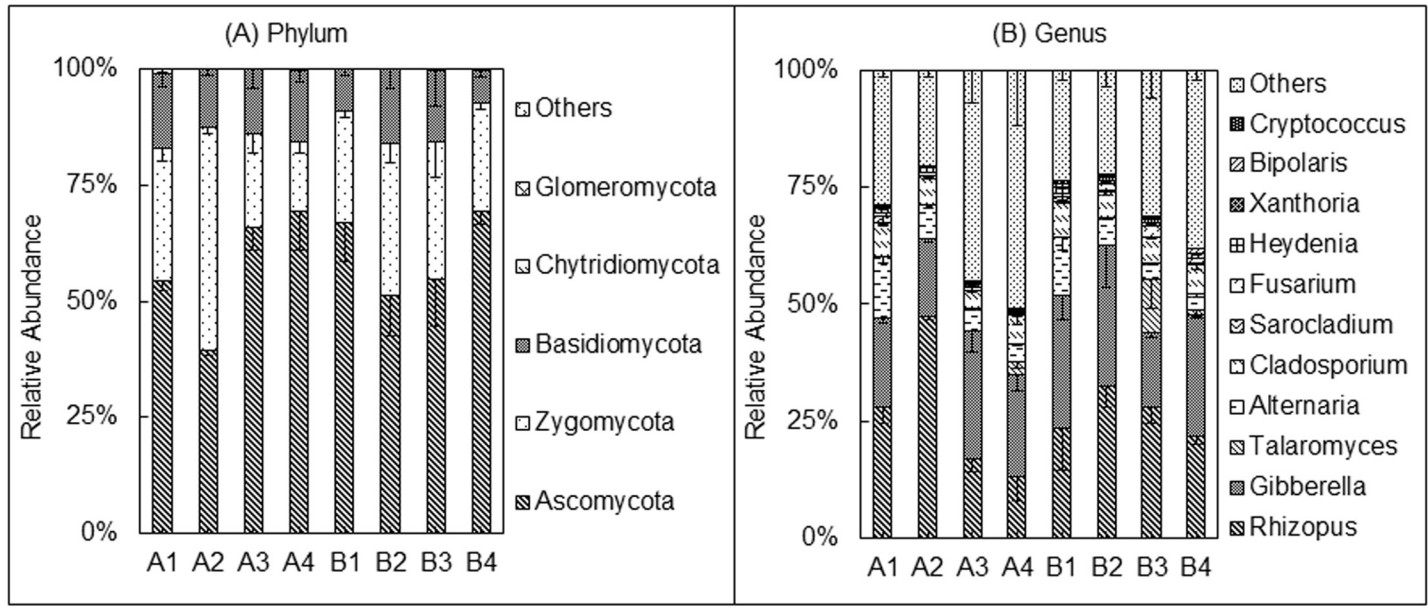

**Fig 3. Comparison of the fungal community structures between different oat cultivars and soil amendment treatments.** A and B represent the Caoyou1 and Baiyan2 oat cultivars, respectively. A1 and B1 represent the negative control; A2 and B2 represent the bio-fertilizer treatment; A3 and B3 represent the rotten straw treatment; and A4 and B4 represent the bio-fertilizer and rotten straw treatment.

Table). There were no significant differences in *Rhizopus* and *Gibberella* abundances between B1 and B4 treatments (Fig 3B and S2 Table).

## Correlation analysis of fungal communities against oat morphologies and exudates

RDA results revealed that the first two axes explain 41.00% of the total variation in fungal OTU matrices (Fig 4). In particular, soluble sugar and total organic acid levels in the rhizosphere were closely positively related to the negative control (treatments A1 and B1) and the bio-fertilizer (treatments A1 and B1) of both cultivars. This contributed significantly to the explanatory power of the RDA model (Fig 4 and S4 Table). Most tested oat morphological parameters (particularly root length, root surface area, and shoot fresh biomass) were

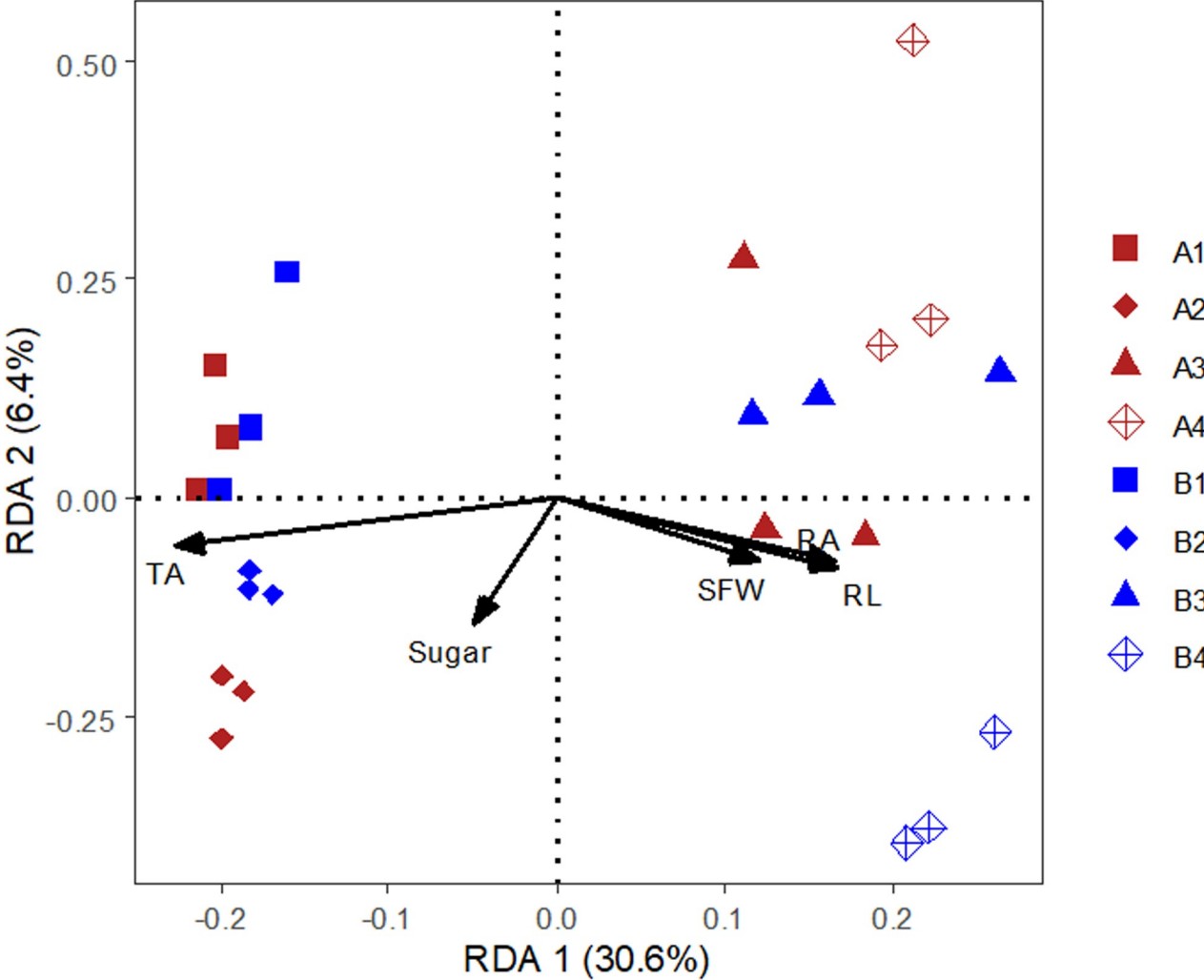

**Fig 4. Redundancy analysis (RDA) of fungal OTUs matrix, oat morphologies, and root exudates in two oat cultivars and multiple soil amendment treatments.** A and B represent the Caoyou1 and Baiyan2 oat cultivars, respectively. A1 and B1 represent the negative control; A2 and B2 represent the bio-fertilizer treatment; A3 and B3 represent the rotten straw treatment; and A4 and B4 represent the bio-fertilizer and rotten straw treatment.The following parameters were used: Sugar, soluble sugar; TA, total organic acid; RL, root length; RA, root surface area; and GFW, shoot fresh biomass.

positively related to the application of rotten straw (treatments A3, A4, B3 and B4; Fig 4 and S4 Table). Of these treatments, the B4 treatment had the greatest positive impact on root morphological parameters (Fig 4 and S4 Table).

At the phylum level, Spearman correlation analysis showed that total organic acid levels in the rhizosphere were significantly positively correlated with the proportion of *Zygomycota* but were significantly negatively correlated with the proportion of *Ascomycota* (S1 Fig). Moreover, these variables were positively correlated with *Rhizopus*, *Alternaria*, *Sarocladium*, and *Heydenia*, and negatively correlated with *Talaromyces*, *Fusarium*, and *Bipolaris* (Fig 5 and S1 Fig). Soluble sugar and total organic acid levels were also negatively correlated with oat root morphologies and biomass (S5 Table).

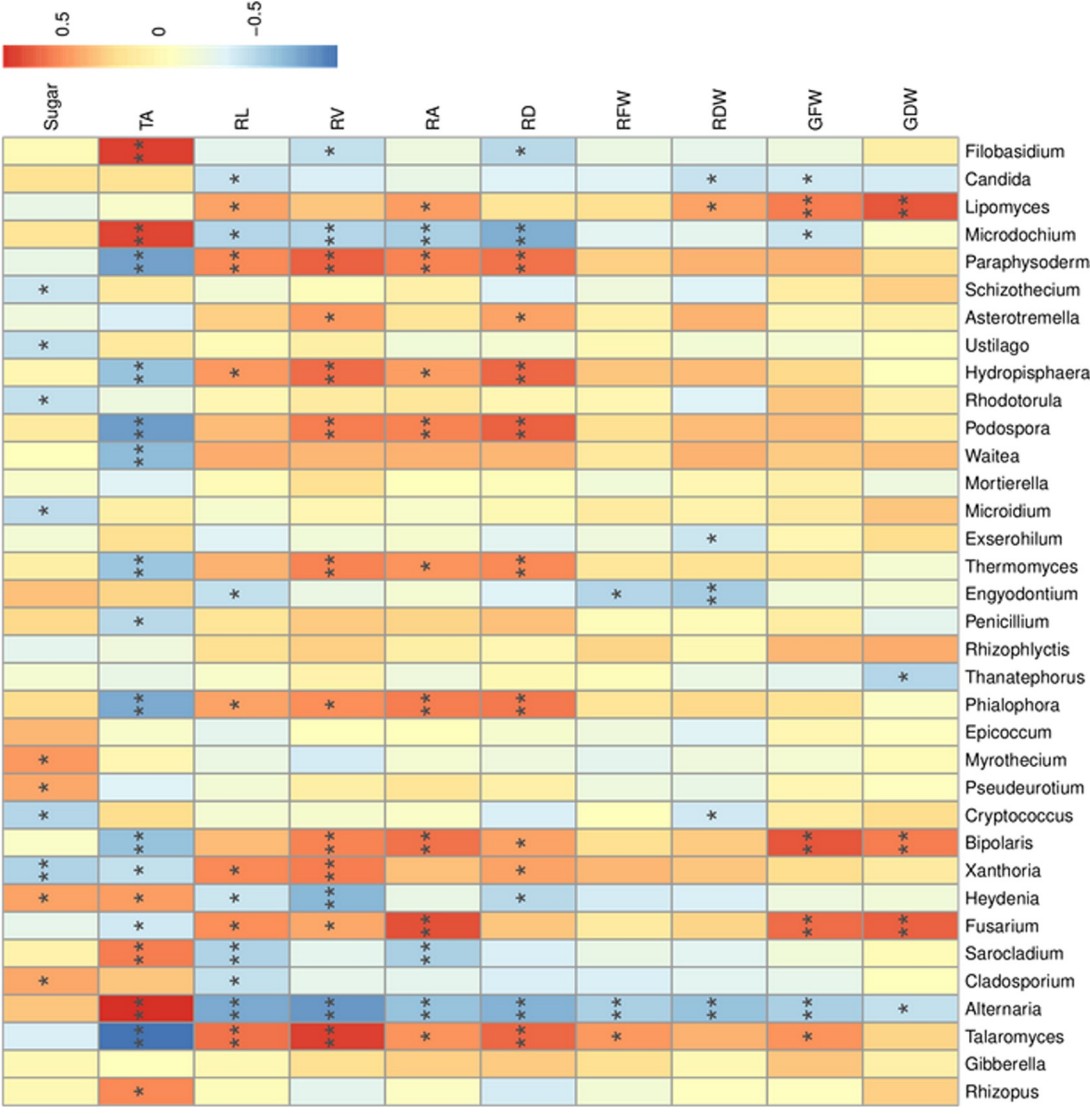

**Fig 5. Spearman's rank-correlations (r) between the top 35 most abundant fungal genera, oat morphologies, and root exudates where the "r" value is between −1 and 1.** An r < 0 indicates a negative correlation and an r > 0 indicates a positive correlation. The labels *, ** and *** indicate significance levels of *p* < 0.05, *p* < 0.01 and *p* < 0.001, respectively. The following parameters were used: Sugar, soluble sugar; TA, total organic acid; RL, root length; RV, root volume; RA, root surface area; RD, root diameter; RFW, root fresh biomass; RDW, root dry biomass; GFW, shoot fresh biomass; and GDW, shoot dry biomass.

## Discussion

### The relationship between oat morphologies and root exudates of two cultivars

This study found that healthy oat morphologies were significantly negative correlated with the root exudates soluble sugar and organic acid under saline-alkaline growing conditions. These findings are consistent with previous research by Boldt-Burisch et al. [20], where root volume was significantly negatively correlated with total amount of organic acids in the rhizosphere both for plants grown in the non-sterilized and sterilized soils. Notably, the tolerant cultivar, Baiyan2, had better plant morphologies and lower levels of soluble sugar and organic acid in the rhizosphere than the sensitive cultivar, Caoyou1. Additionally, the amount of root exudates have been shown to vary between plant species and cultivars under different growing conditions [20–22, 24]. This indicates that the sensitive oat cultivar may secrete more exudates than resistance oat cultivar in response to environmental stress [49], which may cost more plants energy and led to restricted plant growth [23, 24]. Therefore, if root exudates can be used as an indicator of plant health, these compounds not only identify tolerant cultivars but may also be a suitable strategy for increasing crop productivity in saline-alkaline soils.

### The relationships between oat morphologies and root exudates of two cultivars under different amendments

Organic amendments have been shown to optimize nutrient conditions in the soil and promote plant growth [6, 13, 14]. In accordance with these findings, we observed that application of organic soil amendments increased oat root development and biomass. Amendments also had a significant effect on the rhizosphere levels of soluble sugar and organic acids.

Plants secrete root exudates in order to survive in poor nutrient conditions [23–25]. The decreased total organic acid levels observed in the treatments involving rotten straw confirm previous findings that organic acids are not necessary for plant survival in saline-alkaline soils when organic materials are provided [20, 36]. However, bio-fertilizer application significantly increased malic and citric acid content for both cultivars, and combined amendments significantly increased oxalic, acetic, and succinic acid content for Baiyan2 in our study. This could be explained by the findings of previous studies that have reported that bio-fertilizers with live microbes may release certain organic acids [10, 31, 32]. Increased in oxalic, malic, citric acids and so on in root exudates can be acted as chemoattractants of soil microbes during both pathogenic and beneficial interactions [23, 24].

Sugars from roots represent the main source of energy for microorganisms in the rhizosphere, and thus be good for beneficial microorganisms and have a beneficial effect on plant productivity under the better nutrient conditions [35]. The combined bio-fertilizer and rotten straw treatment significantly increased soluble sugar in the rhizosphere of Baiyan2, which may be because higher sugar concentration in roots were released [50]. However, soluble sugar levels were significantly decreased by the bio-fertilizer and rotten straw alone treatments in our study, which may be because the addition of amendments supplied additional carbon source for microorganisms and decreased the uptake of root exudates by organisms [36]. If root exudates respond differently to different amendments, it indicates that an appropriate amendment is worth considering in agricultural systems.

### The relationships between root exudates and fungal communities of two oat cultivars

Previous research has revealed that soil microbial communities, and particularly rhizosphere fungal communities [30], utilize and are influenced by root exudates [27]. In this study, total

organic acid levels had a negative relationship with *Ascomycota* and a positive relationship with *Zygomycota*. Additionally, we found that *Ascomycota* and *Zygomycota* were the dominant fungal phyla in the saline-alkaline environment, which concurs with previous reports from Northeast China [51]. The abundances of *Gibberella*, *Talaromyces*, *Fusarium*, *Heydenia*, and *Bipolaris* were higher in Baiyan2 than Caoyou1, which may be due to the suppressive effects of root exudates that more existed in rhizosphere of Caoyou1 than Baiyan2. This explanation is supported by similar results that organic acids can suppress pathogens and control soilborne diseases from previous studies [52, 53]. In particular, Gomes et al. [54] indicated that cultivar type played a critical role in mediating the fungal community composition. Hence, our study suggested that plants exude organic acids to inhibit potentially harmful microorganisms in the rhizosphere that can cause restricted root development and reduced crop biomass in saline-alkaline soils.

## The relationships between root exudates and fungal communities of two oat cultivars under different amendments

Bio-fertilizer considerably increased the proportion of *Rhizopus arrhizus* in both oat cultivars. *Rhizopus arrhizus* has been widely used for bio-remediation of saline-alkaline soil [33, 34], and belongs to the phylum *Zygomycota*, which contains species that are known to produce organic acids like malic, citric, fumaric acid that can inhibit pathogens [32, 55]. Indeed, the application of bio-fertilizer increased malic, citric and fumaric acid, resulting in higher total organic acid levels, especially in the rhizosphere of Baiyan2, which had a significant positive relationship with *Rhizopus* abundance. It was worthy to say that the bio-fertilizer may act directly on soil nutrients, promoting plant growth, which allows soil microorganisms to receive more organic acids from the host plant [35]. The bio-fertilizer also significantly limited *Ascomycota*, as well as *Alternaria*, *Cladosporium*, *Sarocladium* and *Heydenia*, which are all associated with disease and frequently occur together [54, 56–58]. This may be due to the increased in organic acids in the rhizosphere as a result of the amendment [55]. However, the bio-fertilizer also significantly reduced fungal diversity by increasing organic acid levels. This study confirms that bio-fertilizer application can increase relative organic acid level by promoting the growth of antagonistic organisms or plants, thereby restricting some fungal pathogens and providing a healthy living environment for plants.

The amendments containing rotten straw significantly increased the observed fungal richness and diversity, as well as the abundance of *Ascomycota* (mainly *Gibberella*, *Talaromyces*, *Fusarium* and *Bipolaris*) compared to the control, especially in the rhizosphere of Caoyou1. This is due to the decrease in soluble sugar and organic acids in the rhizosphere of Caoyou1 caused by the rotten straw application. Fungal genera *Gibberella*, *Fusarium*, and *Bipolaris*, can be harbored in soils [59] and are common pathogens that lead to crop yield loss [52, 60, 61]. The amendments containing rotten straw significantly decreased *R. arrhizus* for the sensitive cultivar Caoyou1, but not for the tolerant cultivar Baiyan2. This may not only because the organic acids were decreased by the rotten straw application in Caoyou1, but also because more carbohydrates were supplied by the strong roots of the resistant cultivar Baiyan2 under the better nutrient conditions [35], especially in the combined bio-fertilizer and rotten straw treatment. This suggests that rhizosphere fungal communities in saline-alkaline tolerant oat cultivars are protected and promoted in poor soil conditions if we consider the co-application of bio-fertilizer and rotten straw as the opportune method, because the developed oat roots can secrete root exudates to maintain the fungal communities in saline-alkaline environments.

## Conclusions

The evidence presented here indicates that root exudate levels can be used to evaluate the relationships between oat plants and rhizosphere fungal communities in saline-alkaline

environments. Our study confirmed that bio-fertilizers decrease fungal diversity and reduce harmful pathogens by promoting oat root and *Rhizopus arrhizus* organic acid secretions. Conversely, we showed that that the amendments containing rotten straw increased fungal richness, diversity, and pathogenic fungi due to a reduction in root exudates. The combined bio-fertilizer and rotten straw amendment significantly improved oat root development and overall biomass by promoting root exudates that stabilized the rhizosphere fungal communities in the tolerant oat cultivar, Baiyan2. We suggest that co-application of a bio-fertilizer and rotten straw, along with a tolerant cultivar, is an effective method to improve crop growth and enhance soil health in saline-alkaline environments, which can be adopted to improve crop productivity. In addition, we found a negative relationship between *R. arrhizus* and several pathogenic fungi. Thus, further studies on *R. arrhizus* as a potential biocontrol agent should be conducted.

## Supporting information

**S1 Fig. Spearman's rank-phylum correlations (r) between the top 10 most abundant fungal phylum, oat characteristics, and root exudates where "r" value is between −1 and 1.** An $r < 0$ indicates a negative correlation and an $r > 0$ indicates a positive correlation. The labels *, ** and *** indicate significance levels of $p < 0.05$, $p < 0.01$ and $p < 0.001$, respectively. The following parameters were used: sugar, soluble sugar; TA, total organic acid; RL, root length; RV, root volume; RA, root surface area; RD, root diameter; RFW, root fresh biomass; RDW, root dry biomass; GFW, shoot fresh biomass; and GDW, shoot dry biomass.
(TIF)

**S1 Table. Effects of amendments on fungal community compositions of two oat cultivars at the phylum level.**
(XLSX)

**S2 Table. Effects of amendments on fungal community compositions of two oat cultivars at the genus level.**
(XLSX)

**S3 Table. Effects of amendments on fungal community compositions of two oat cultivars at the species level.**
(XLSX)

**S4 Table. Proportion of variance explained by environmental variables as determined by RDA for fungal OTUs matrix.**
(XLSX)

**S5 Table. Pearson correlation analysis between oat morphological characters and root exudates in two oat cultivars and multiple soil amendment treatments (* P < 0.05, ** P < 0.01, *** P < 0.001).**
(XLSX)

## Acknowledgments

We are very grateful to Inner Mongolia Agriculture, Animal Husbandry, Fishery and Biology Experiment Research Centre for organic acids analysis and Novogene Co., Ltd. for ITS sequence barcode pyrosequencing. We thank Dr. Madeleine Thomson for modifying and proofreading the manuscript.

## Author Contributions

**Conceptualization:** Baoping Zhao, Jinghui Liu.

**Data curation:** Peina Lu.

**Formal analysis:** Peina Lu.

**Funding acquisition:** Lijun Li, Baoping Zhao, Jinghui Liu.

**Investigation:** Lijun Li, Baoping Zhao, Jinghui Liu.

**Methodology:** Lijun Li, Baoping Zhao, Jinghui Liu.

**Project administration:** Lijun Li, Jinghui Liu.

**Resources:** Jinghui Liu.

**Software:** Peina Lu.

**Supervision:** Jinghui Liu.

**Validation:** Peina Lu.

**Visualization:** Peina Lu.

**Writing – original draft:** Peina Lu.

**Writing – review & editing:** Peina Lu, Tony Yang, Jinghui Liu.

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
