## [Decision Letter · Decision Letter 0]

21 Sep 2020

PONE-D-20-18342

Response of oat morphologies, root exudates, and rhizosphere fungal communities to amendments in a saline-alkaline environment

PLOS ONE

Dear Dr. Liu,

Thank you for submitting your manuscript to PLOS ONE. After careful consideration, we feel that it has merit but does not fully meet PLOS ONE’s publication criteria as it currently stands. Therefore, we invite you to submit a revised version of the manuscript that addresses the points raised during the review process.

I would like you to focus particularly on the issues of presenting the results with more accuracy without any contradictions and improvement in the Discussion section.

We look forward to receiving your revised manuscript.

Kind regards,

Birinchi Sarma, PhD

Academic Editor

PLOS ONE

Journal Requirements:

2.We note that you have indicated that data from this study are available upon request. PLOS only allows data to be available upon request if there are legal or ethical restrictions on sharing data publicly. For more information on unacceptable data access restrictions, please see http://journals.plos.org/plosone/s/data-availability#loc-unacceptable-data-access-restrictions.

Reviewers' comments:

Reviewer's Responses to Questions

**Comments to the Author**

1. Is the manuscript technically sound, and do the data support the conclusions?

Reviewer #1: Yes

Reviewer #2: Yes

2. Has the statistical analysis been performed appropriately and rigorously? 

Reviewer #1: Yes

Reviewer #2: Yes

3. Have the authors made all data underlying the findings in their manuscript fully available?

Reviewer #1: Yes

Reviewer #2: Yes

4. Is the manuscript presented in an intelligible fashion and written in standard English?

Reviewer #1: Yes

Reviewer #2: Yes

5. Review Comments to the Author

Reviewer #1: The topic is on the application of organic amendments to saline-alkaline soil recommended

as an agricultural strategy to improve crop productivity and soil health.

The manuscript is well presented;however, some details must be checked and corrected:

FIGURE 3. (A) iT CAN BE IMPROVED. Not to use colours and maybe to use different patterns.

To check for all figures presentation.

Reviewer #2: The authors described some interesting observation of soil amendments on root exudates and fungal communities in oat plants under saline alkaline conditions. Results from the study show that combined soil application of bio-fertilizer and rotten straw on tolerant Baiyan2 oat cultivar can be effective improving oat morphologies, soluble sugar in rhizosphere in a saline-alkaline environment. The bio-fertilizer amendment also significantly limited disease causing fungi.

Comments:

Materials and Methods: After how many days of soil amendments were the samples collected?

Results: Line 228-229: As per Table 1, a significant change in shoot dry biomass was observed between the oat cultivars (1.9b and 3.4a; different letters indicate significance). However, the text contradicts the result in the table and says no significant change in shoot dry biomass.

Line 267-268: Caoyou1 needs to be added "All tested organic acids except fumaric acid were

significantly reduced in treatments containing rotten straw (R and RF; Table 2) in Caoyou1".

Though the manuscript had necessary comparisons of their results with published results but the entire discussion lacks a quantitative description of each phenomenon. For e.g. Increase in a particular organic acid in root exudate can be explained along with their possible role. In addition, the authors state in their conclusion that "that co-application of a bio-fertilizer and rotten straw, along with a tolerant cultivar, is an effective method to improve crop productivity in saline-alkaline environments. Was any yield/productivity measured? If yes they can add the data or should say that this method can be adopted to improve crop productivity.

Minor corrections:

Line 77: Space between "often" and "affected"

Line 80-83: Reframe line 80-83

Line 82- Replace "benefit" with "beneficial"

Line 83-Space between "acid" and "in" in "acidin"

LINE 215: Expand ANOVA as analysis of variance

Line 261: Replace"Cayou1" with "Caoyou1"

Line 391- Replace "negative" with "negatively"

Line 391: Add "where" before "root"

Line 392: Rewrite the part “whatever plants grown in the non-sterilized and sterilized soils".

Line 398: Remove “to” in between "restricted to plant growth"

Line 425- "indicated" to be replaced with "indicates"

Line 459: Replace "related" with "relative"

Line 462: Replace "fungal observed richness" with "observed fungal richness"

Line 469: Replace "decreased" with "decrease"

Line 468-472: Reframe the sentence and check the need to write "not" between "could" and "prevent"

Figure 1: x-axis legend "e" missing from root volume

6. PLOS authors have the option to publish the peer review history of their article (what does this mean?). If published, this will include your full peer review and any attached files.

Reviewer #1: **Yes: **Marcela Pagano

Reviewer #2: **Yes: **Akansha Jain

---

## [Author Response · Author response to Decision Letter 0]

31 Oct 2020

Below are our point-by-point responses to the editor's and referees' comments: The reviewers and editor’s comments are given in blue font, our responses in black font.

Reviewer #1: The topic is on the application of organic amendments to saline-alkaline soil recommended

as an agricultural strategy to improve crop productivity and soil health.

Comments:

The manuscript is well presented; however, some details must be checked and corrected:

FIGURE 3. (A) iT CAN BE IMPROVED. Not to use colours and maybe to use different patterns.

To check for all figures presentation.

Response: Thanks for your comments. According to your suggestions, we have changed the colors to the different patterns in Figure 3. We also have checked all figures for presentation. 

Reviewer #2: The authors described some interesting observation of soil amendments on root exudates and fungal communities in oat plants under saline alkaline conditions. Results from the study show that combined soil application of bio-fertilizer and rotten straw on tolerant Baiyan2 oat cultivar can be effective improving oat morphologies, soluble sugar in rhizosphere in a saline-alkaline environment. The bio-fertilizer amendment also significantly limited disease causing by fungi.

Comments:

Materials and Methods: After how many days of soil amendments were the samples collected?

Results: Line 228-229: As per Table 1, a significant change in shoot dry biomass was observed between the oat cultivars (1.9b and 3.4a; different letters indicate significance). However, the text contradicts the result in the table and says no significant change in shoot dry biomass.

Line 267-268: Caoyou1 needs to be added "All tested organic acids except fumaric acid were

significantly reduced in treatments containing rotten straw (R and RF; Table 2) in Caoyou1".

Though the manuscript had necessary comparisons of their results with published results but the entire discussion lacks a quantitative description of each phenomenon. For e.g. Increase in a particular organic acid in root exudate can be explained along with their possible role. In addition, the authors state in their conclusion that "that co-application of a bio-fertilizer and rotten straw, along with a tolerant cultivar, is an effective method to improve crop productivity in saline-alkaline environments. Was any yield/productivity measured? If yes they can add the data or should say that this method can be adopted to improve crop productivity.

Response: Thanks for your comments. First, we have added the sample collection time in Materials and Methods part (Line 139). Second, we have corrected the “shoot dry biomass” to “root dry biomass”(Line 230), and we have added “in Caoyou1”(Line 269-270) in Results part. Third, we have added the sentence “Increased in oxalic, malic, citric acids and so on in root exudates can be acted as chemoattractants of soil microbes during both pathogenic and beneficial interactions [1, 2].” in Discussion part (Line 417-419) to make it more clear. Fourth, we have corrected the “productivity” to the “growth” and have added “, which can be adopted to improve crop productivity” in order to make it more clear here in Conclusions part (Line 494-495). 

1. Lombardi N, Vitale S, Turra D, Reverberi M, Fanelli C, Vinale F, et al. Root exudates of stressed plants stimulate and attract trichoderma soil fungi. Mol Plant Microbe Interact. 2018;31(10):982-94.

2. Huang X, Chaparro JM, Reardon KF, Zhang R, Shen Q, Vivanco JM. Rhizosphere interactions: root exudates, microbes, and microbial communities. Botany. 2014;92(4):267-75.

Minor corrections:

Line 77: Space between "often" and "affected"

Response: We have corrected it.

Line 80-83: Reframe line 80-83

Response: We have reframed this sentences.

Line 82- Replace "benefit" with "beneficial"

Response: We have corrected it.

Line 83-Space between "acid" and "in" in "acidin"

Response: We have corrected it.

LINE 215: Expand ANOVA as analysis of variance

Response: We have corrected it.

Line 261: Replace"Cayou1" with "Caoyou1"

Response: We have modified it.

Line 391- Replace "negative" with "negatively"

Response: We have modified it.

Line 391: Add "where" before "root"

Response: We have modified it.

Line 392: Rewrite the part “whatever plants grown in the non-sterilized and sterilized soils".

Response: We have modified it.

Line 398: Remove “to” in between "restricted to plant growth"

Response: We have modified it.

Line 425- "indicated" to be replaced with "indicates"

Response: We have modified it.

Line 459: Replace "related" with "relative"

Response: We have modified it.

Line 462: Replace "fungal observed richness" with "observed fungal richness"

Response: We have modified it.

Line 469: Replace "decreased" with "decrease"

Response: We have modified it.

Line 468-472: Reframe the sentence and check the need to write "not" between "could" and "prevent"

Response: We have modified it.

Figure 1: x-axis legend "e" missing from root volume

Response: We have modified it.

Thanks again.

Sincerely yours,

Dr. Jinghui Liu

Professor of Agronomy

---

## [Editor Report · Decision Letter 1]

19 Nov 2020

Response of oat morphologies, root exudates, and rhizosphere fungal communities to amendments in a saline-alkaline environment

PONE-D-20-18342R1

Dear Dr. Liu,

We’re pleased to inform you that your manuscript has been judged scientifically suitable for publication and will be formally accepted for publication once it meets all outstanding technical requirements.

Kind regards,

Birinchi Sarma, PhD

Academic Editor

PLOS ONE
---

## [Editor Report · Acceptance letter]

24 Nov 2020

PONE-D-20-18342R1 

Response of oat morphologies, root exudates, and rhizosphere fungal communities to amendments in a saline-alkaline environment 

Dear Dr. Liu:

I'm pleased to inform you that your manuscript has been deemed suitable for publication in PLOS ONE. Congratulations! Your manuscript is now with our production department. 

Kind regards, 

on behalf of

Dr. Birinchi Sarma 

Academic Editor

PLOS ONE